# Association of CD47 Expression with Clinicopathologic Characteristics and Survival Outcomes in Muscle Invasive Bladder Cancer

**DOI:** 10.3390/jpm13060885

**Published:** 2023-05-24

**Authors:** Zin W. Myint, Zena Chahine, Rani Jayswal, Emily Bachert, Robert J. McDonald, Stephen E. Strup, Andrew C. James, Patrick J. Hensley, Derek B. Allison

**Affiliations:** 1Division of Medical Oncology, Department of Internal Medicine, University of Kentucky, Lexington, KY 40536, USA; 2Markey Cancer Center, University of Kentucky, Lexington, KY 40536, USA; 3Department of Pathology and Laboratory Medicine, University of Kentucky, Lexington, KY 40536, USA; 4Department of Urology, University of Kentucky, Lexington, KY 40536, USA

**Keywords:** muscle invasive bladder cancer, CD47 expression, neoadjuvant therapy, biomarker

## Abstract

**Simple Summary:**

CD47 is a transmembrane protein expressed at a basal level in many cell types but is often overexpressed in tumor cells. CD47 overexpression has been correlated with adverse clinical outcomes in several malignancies. Hence, CD47 could be a promising candidate for target therapy in future cancer treatment. In this retrospective study of 87 patients with muscle invasion bladder cancer (MIBC), we examined CD47 IHC expressions in tumor samples from transurethral resections of bladder tumors (TURBT) and matched radical cystectomy (RC) specimens. We found detectable CD47 expressions in 44% of TURBT samples, but it was not a predictive or prognostic marker for MIBC patients. However, in patients receiving neoadjuvant chemotherapy (NAC), there was a positive trend toward decreased CD47 levels from TURBT to RC. The study suggests that further research is needed to understand the potential role of anti-CD47 therapy in MIBC patients and how NAC may modify immune surveillance mechanisms.

**Abstract:**

Objective: CD47 is an antiphagocytic molecule that plays a critical role in immune surveillance. A variety of malignancies have been shown to evade the immune system by increasing the expression of CD47 on the cell surface. As a result, anti-CD47 therapy is under clinical investigation for a subset of these tumors. Interestingly, CD47 overexpression is associated with negative clinical outcomes in lung and gastric cancers; however, the expression and functional significance of CD47 in bladder cancer is not fully understood. Materials and Methods: We retrospectively studied patients with muscle invasion bladder cancer (MIBC) who underwent a transurethral resection of bladder tumor (TURBT) and subsequently underwent radical cystectomy (RC) with or without neoadjuvant chemotherapy (NAC). CD47 expression was examined by IHC in both TURBT and matched RC specimens. The difference in CD47 expression levels between TURBT and RC was also compared. The association of CD47 levels (TURBT) with clinicopathological parameters and survival outcomes was evaluated by Pearson’s chi-squared tests and the Kaplan–Meier method, respectively. Results: A total of 87 MIBC patients were included. The median age was 66 (39–84) years. Most patients were Caucasian (95%), male (79%), and aged >60 (63%) and most often (75%) underwent NAC prior to RC. Of those who received NAC, 35.6% were responders and 64.4% were non-responders. The final reported stages as per AJCC for all patients were as follows: stage 0 (32%), stage 1 (1%), stage 2 (20%), stage 3 (43%), and stage 4a (5%). A total of 60% of patients were alive; of those, 30% had disease recurrence and 40% died from bladder cancer at a median follow-up of 3.1 (0.2–14.2) years. CD47 levels were detectable in 38 (44%) TURBT samples. There was no association between CD47 levels and clinicopathological parameters such as age, gender, race, NAC, final stage, disease recurrence, and overall survival (OS). Patients aged >60 (*p* = 0.006), non-responders (*p* = 0.002), and at stage ≥ 3 (*p* < 0.001) were associated with worse OS by a univariate analysis and stage ≥ 3 remained significant even after a multivariate analysis. In patients managed with NAC, there were decreased CD47 levels in RC specimens compared to the TURBT specimens, but this did not reach statistical significance. Conclusion: CD47 expression was not a predictive nor prognostic marker for MIBC patients. However, expression of CD47 was detected in nearly half of MIBCs, and future studies are needed to explore the potential role of anti-CD47 therapy in these patients. Furthermore, there was a slight positive trend in decreased CD47 levels (from TURBT to RC) in patients receiving NAC. As a result, more research is needed to understand how NAC may modify immune surveillance mechanisms in MIBC.

## 1. Introduction

CD47 is a transmembrane receptor that results in the inhibition of phagocytosis by macrophages, serving as a “don’t eat me” signal. Although ubiquitously expressed at low levels, a variety of malignancies show increased expression of CD47 on the cell surface [1,2]. More specifically, CD47 overexpression has been correlated with adverse clinical outcomes in several malignancies, such as esophageal cancer [3], ovarian cancer [4], glioblastoma [5], hematologic malignancies [6,7], and breast cancer [8]. Although some preliminary data suggest that bladder cancer shows increased expression of CD47 [1,9,10,11], much remains unknown about its clinicopathologic significance in muscle invasive bladder cancer (MIBC) and its role in targeted therapeutic strategies. However, based on limited preliminary data, it is worth further exploration. For example, blocking CD47 with a monoclonal antibody was shown to enable macrophages to phagocytose bladder cancer cells in vivo, inhibit tumor growth, and prevent metastases in xenotransplantation models [1,11]. Based on similar data from other tumors, there are several ongoing early clinical trials on hematologic and solid malignancies. Unfortunately, studies utilizing human clinical bladder cancer samples to study CD47 expression are limited.

Since >80% of bladder cancer mortality is from muscle invasive disease, more studies are warranted in this cohort. The reference standard treatment for MIBC is neoadjuvant chemotherapy (NAC) prior to radical cystectomy (RC). Studying this population can allow investigation into how CD47 expression predicts the response to platinum-based therapy, which has not yet been studied, and could help determine if there is a rational basis for combination or adjuvant anti-CD47 therapy as a future direction. The following study was performed to determine the association of CD47 expression with clinicopathologic characteristics and survival outcomes in a clinical cohort of patients with MIBC managed with and without cisplatin-based NAC followed by RC.

## 2. Materials and Methods

This study is retrospective and was conducted according to the guidelines of the Declaration of Helsinki and approved by the local Institutional Review Boards at the University of Kentucky. The approval number is 54170.

Patients with MIBC diagnosed and treated at our institution between April 2007 and July 2016 were included in this study. More specifically, the inclusion criteria were as follows: ≥18 years of age, conventional high-grade urothelial carcinoma, clinical stage T2, and above on TURBT (transurethral resection of bladder tumor) with subsequent RC with or without NAC. The exclusion criteria were cases with no available tissues and patients who underwent chemoradiation treatment or partial cystectomy. NAC regimens were cisplatin based. The final pathology was stratified into responders vs. non-responders. Responders were defined as those with a pathologic complete response (ypT0N0) or a partial response (ypTis or ypT1N0) and non-responders were those with stage ≥ ypT2 and or any ypTN+. Data on age, gender, race, NAC, surgery, final stage, tumor recurrence, and mortality status were collected through the Kentucky Cancer Registry (KCR) database. Tumor recurrence is defined as either local recurrence or any distant metastasis.

### 2.1. CD47 Immunohistochemistry

CD47 expression was examined by immunohistochemistry (IHC) in both TURBT and matched RC specimens. First, formalin-fixed, paraffin-embedded (FFPE) tissue blocks were cut at four micron thickness and baked for a minimum of one hour at 58 °C. IHC staining was conducted utilizing the Ventana Discovery Ultra platform with Standard CC2 (Roche, Tucson, AZ, USA) antigen retrieval, which contains a citrate/low pH buffer, at 95 °C for 1 h. The tissue was then incubated with an anti-CD47 antibody (ab226837, Abcam, Cambridge, MA, USA) at 1:50 dilution for 1 h at room temperature, followed by 20 min of incubation with OmniMap anti-Rabbit-HRP (Roche, Tucson, AZ, USA) with visualization using DAB (Roche), according to the manufacturer’s recommendations. The staining intensity was scored as follows: 0 = no staining, 1 = weak staining, 2 = moderate staining, and 3 = strong staining (Figure 1). Tumor percentage staining was scored as follows: 0 = 0%, 1 = 1–5%, 2 = 5–20%, 3 = 20–50%, and 4 = 50–100%. A composite score for each sample was calculated by multiplying the staining intensity score by the tumor percentage staining score, resulting in a range of possible scores from 0 to 12. Interpretation was performed by two board-certified anatomic pathologists (authors RJM and DBA). After analyzing the distribution of staining, cases showing CD47 expression (TURBT specimens) were considered positive for evaluation. These results were then compared to clinicopathological parameters and survival outcomes. Additionally, the difference in CD47 expression levels between TURBT and RC were also compared. Based on the CD47 IHC changes, we defined expression as “stable” vs. “up” vs. “down”. For example, if CD47 IHC from the TURBT was scored as 12 and the CD47 IHC from the matched RC was scored as 4, then it fell into the category of “down”. We wanted to see whether CD47 expression scoring changes had any association with the response to platinum-based chemotherapy.

### 2.2. Statistical Analysis

A total of 87 patients were included in the analysis. Basic characteristics including age, gender, race, NAC, pathological stage, final stage, tumor recurrence, survival status, and CD47 expression were summarized by descriptive statistics. Univariate and multivariate analyses were performed by using Cox proportional hazard models, where the Hazard Ratio (HR) and their 95% confidence intervals (95% CI) were reported. The chi-square test was used to compare each group for categorical variables. PFS and OS were estimated using the Kaplan–Meier method, and comparisons among survival times were analyzed with a log-rank test. To study the differences in CD47 IHC expression levels between TURBT specimens and post-cystectomy specimens, Wilcoxon signed rank tests were performed. Comparisons between survival distributions were performed by the log-rank test. A two-sided *p* value of <0.05 was considered statistically significant. Data were analyzed with SAS 9.4 (SAS Institute, Cary, NC, USA).

## 3. Results

### 3.1. Baseline Characteristics

A total of 87 MIBC patients were included. The median age was 66 (39–84) years. Most patients were Caucasian (95%), male (79%), and aged >60 (63%). All patients presented with Eastern Cooperative Oncology Group (ECOG) performance status < 2 at the time of diagnosis (100%). Most patients (75%) underwent NAC prior to RC. Of those who received NAC, 35.6% were responders and 64.4% were non-responders. The definitions of responders and non-responders were described in the methods section. The final reported stages for all patients were as follows: stage 0 (32%), stage 1 (1%), stage 2 (20%), stage 3 (43%), and stage 4a (5%). A total of 60% of patients were alive; of those, 30% had disease recurrence and 40% died from bladder cancer at a median follow-up of 3.1 (0.2–14.2) years. CD47 levels were detectable in 38 (44%) TURBT samples and 15 (38%) RC samples.

### 3.2. Association of CD47 Expression with Clinical Parameters

The mean CD47 expression score for TURBT samples was 1.59, and the standard deviation was 2.7. Forty-nine samples showed no expression, thirty samples had a score between 1 and 4, five samples had a score between 5 and 8, and three samples had a score between 8 and 12 (Figure 2). Due to this distribution of staining, a dichotomous definition of negative (expression score 0) and positive (expression score ≥ 1) was used for statistical purposes. Based on this definition of CD47 scoring in TURBT specimens, 87 patients were stratified into two groups. The CD47 negative group (n = 49) vs. CD47 positive group (n = 38) were analyzed against various clinicopathologic parameters. There was no association between the CD47 level (TURBT samples) and age, gender, race, NAC, final stage, disease recurrence, and overall survival (OS) (Table 1).

### 3.3. Survival Outcomes

The median follow-up of patients was 3.9 years (0.2–14.2 years). The median progression-free survival (PFS) was 5 years (0.06–8.7 years) and the median overall survival (OS) was 5.7 years (0.2–14.2 years). The median OS rate for the patients with CD47 positive cases was lower than that for the CD47 negative cases, at 4.1 years vs. 5.9 years, respectively (*p* = 0.7). Table 2 shows the univariate and multivariate analyses of clinicopathological factors associated with PFS and OS. Patients > 60, non-responders, and stage ≥ 3 were associated with worse PFS and OS by a univariate analysis. When adjusted to stage by a multivariate analysis, only stage ≥ 3 was independently associated with survival (*p* < 0.001) (Figure 3A). In our cohort, there was no association between CD47 expression and survival (Figure 3B).

### 3.4. Additional Analysis

We further analyzed CD47 IHC changes in patients who had matched tissue available from TURBT and RC samples; as a result, patients who underwent a complete pathologic response with no tumor identified on RC were not studied due to a lack of post-treatment tumor. A total of 40 patients were included. Based on the CD47 IHC changes, we defined the expression as “stable” vs. “up” vs. “down”. Table 3 shows the association between NAC (responders vs. non-responders) and changes in CD47 IHC scoring (from TURBT and matched RC samples). There was a slight positive trend for decreased CD47 levels between TURBT and RC in patients who received NAC (*p* = 0.5), though this did not reach statistical significance (Figure 4).

## 4. Discussion

CD47 is an immunoglobulin-like transmembrane receptor that has several diverse physiologic functions, including cell migration, T and dendritic cell activation, and axon development [12,13,14,15]. Importantly, it also functions as an immunosuppressive signaling molecule as an inhibitor of phagocytosis, which allows malignant cells to evade destruction by macrophages when the expression is increased [7]. Briefly, the CD47 cell surface receptor on tumor cells forms a complex with signal-regulatory protein alpha (SIRPα) on phagocytic cells, resulting in the recruitment of SHP-1 and SHP-2 that prevent the accumulation of myosin IIA in the phagocytic synapse, ultimately inhibiting phagocytosis [16,17,18]. Some have referred to this function as a “don’t eat me” signal, which is similar but different to the function of PD-L1, providing a “don’t see me” signal. Several human cancers have been shown to evade the immune system by upregulating CD47 on the cell surface and exploiting this mechanism that functionally prevents the unnecessary destruction of normal “self” cells [19].

However, the mechanism of CD47 expression in cancer tissues is complex and appears to go beyond simply a “don’t eat me” signal. For example, CD47 also interacts with thrombospondin-1 (TSP-1), which is a potent inhibitor of angiogenesis [20]. Interestingly, Chan et al. identified an association between the gene expression of CD47 and cancer stem cells, which are thought to be responsible for tumor initiation and disease progression [11]. Their data suggest that the correlation between CD47 and SIRPα expression may play a key role in the progression of bladder cancer [11]. Subsequently, this same group showed that blocking CD47 with a monoclonal antibody enabled macrophages to phagocytose bladder cancer cells in vivo, inhibit tumor growth, and prevent metastases in xenotransplantation models [1,11]. As a result, CD47 could be a promising candidate for target therapy in future cancer treatments. Subsequently, Pan et al. performed endoscopic molecular imaging using a CD47 antibody to improve cystoscopic cancer detection and enable image-guided TURBT [21]. In addition, an interesting study by Kiss et al. utilizing cell cultures showed that CD47-targteted near-infrared photoimmunotherapy resulted in increased phagocytosis and cancer cell death [22]. When studied in a murine patient-derived xenograft model, it inhibited tumor growth and improved survival. Taken together, these basic and translational studies are promising, but additional data are needed in human samples to understand the clinical relevance.

Importantly, our data show CD47 is expressed in a significant number (44%) of MIBC patients and can be detected using IHC on TURBT clinical samples. Although there were no associations between CD47 expression and clinicopathological parameters such as age, gender, race, NAC, final stage, disease recurrence, and OS, CD47 may still be a promising predictive marker for the response to targeted therapy, much like PD-L1. Interestingly, there was a slight positive trend for decreased CD47 levels between TURBT and RC in patients who received NAC (*p* = 0.5), though this did not reach statistical significance. However, this finding, overall, suggests that platinum-based chemotherapy does not play a significant role in disrupting any CD47-related immune evasion mechanisms that may be at play. Furthermore, this finding suggests a rational basis for combination therapy to attack tumors with two separate mechanisms of action. However, further investigation is required.

Magrolimab is an anti-CD47 monoclonal antibody that is currently under investigation in clinical trials in refractory indolent B-cell malignancies (NCT04599634) and myeloid malignancies (NCT04778410) and in a phase 2 clinical trial in patients with solid tumors, including metastatic urothelial cancer (NCT04827576). The rational basis for including bladder cancer in a clinical trial is largely based on these preclinical studies that are limited to cell lines, which may not be directly translatable to bladder cancer as a whole, which is known to be remarkably heterogeneous. Very few studies report the expression of CD47 in human clinical samples of bladder cancer.

To date, the mechanism of action regarding CD47 blockade as a treatment strategy goes beyond the reactivation of a phagocytic pathway [2]. For example, antigen presentation to CD4+ and CD8+ T helper and cytotoxic T cells may facilitate an adaptive immune response against tumor neoantigens. This mechanism is promising given the fact that bladder cancer has one of the highest tumor mutational burdens among solid malignancies and has been treated with immunotherapy (i.e., intravesicular BCG) for decades [23]. In addition, with anti-CD47 therapy, tumor cells may be eliminated through natural-killer-cell-mediated antibody-dependent cytotoxicity, as well as complement-dependent cytotoxicity [2]. Finally, anti-CD47 antibodies can stimulate tumor cell apoptosis through a caspase-independent mechanism [19]. Importantly, despite the widespread expression of CD47 in normal tissue throughout the body, preclinical models have shown tolerance to treatment even at high doses [24]—most likely because normal cells do not have prophagocytic signals. However, aging red blood cells do gradually lose CD47 expression and acquire prophagocytic signals that facilitate their physiologic removal within the spleen [25]. As a result, anemia monitoring and mitigation will be an important part of anti-CD47 treatment.

The identification of predictive markers for MIBC has been challenging and has produced some conflicting results [26,27,28,29,30,31]. Furthermore, there has been very little development in terms of novel therapeutic strategies beyond platinum-based NAC. These data indicate CD47 is commonly expressed in MIBC and can be detected in clinical TURBTs and cystectomy specimens by IHC. These findings are important and contribute to a growing body of evidence that CD47 needs to be further investigated in select patients with urothelial carcinoma.

Limitations:

Only primary, conventional muscle invasive bladder cancer samples were included in this study. As a result, no conclusions can be made regarding the role of CD47 in other stages of disease, such as early tumorigenesis or in the metastatic setting. In addition, this study was retrospective in nature. Furthermore, the distribution of staining in cases was mainly limited to low expression with an expression score between 1 and 4. As a result, all data analyses were performed in a dichotomous fashion, which may cause us to miss nuances in expression.

Conclusions:

CD47 expression is not a prognostic marker for MIBC patients. However, expression of CD47 was detected in nearly half of MIBCs, and future studies are needed to explore the potential role of anti-CD47 therapy in these patients. Furthermore, there was a slight positive trend of decreased CD47 levels (from TURBT to RC) in patients receiving NAC. More research is needed to understand how NAC may or may not modify immune surveillance mechanisms in MIBC.

Future Directions:

Kiss et al. studied the impact of macrophage-mediated phagocytosis and anti-tumor activity on bladder cancer cells (639 V) and xenograft models [32]. They investigated magrolimab (anti-CD47 therapy) as monotherapy and in combination with cytotoxic chemotherapy (gemcitabine-cisplatin) both in vivo and in vitro [32]. The study confirmed that combining magrolimab with cytotoxic chemotherapy resulted in synergistic effects, notably enhancing phagocytosis and reducing tumor growth compared to magrolimab monotherapy or chemotherapy alone [32]. Currently, a phase I clinical trial (NCT05738161) is underway to explore the combination of magrolimab with cytotoxic chemotherapy in advanced urothelial carcinoma.

Lakhani et al. conducted a phase 1b study of combined magrolimab and avelumab (immune checkpoint inhibitor) in patients with immunotherapy naïve ovarian cancer who progressed within 6 months of prior platinum-based chemotherapy [33]. The combination was well-tolerated, with an acceptable safety profile, and the recommended phase 2 dose for magrolimab was 45 mg/kg once weekly [33]. There was no reported dose-limiting toxicity. The most common treatment-related adverse events were headaches (62%), fatigue (47%), infusion-related reactions (44%), fever (38%), chills (35%), and nausea (35%). Among those 18 ovarian cancer patients included in the phase 1 study, 56% had stable disease and 44% had disease progression [33].

Based on our own study and the above findings, a future clinical trial design could consider a potential neoadjuvant study that explores the combination of magrolimab and cytotoxic chemotherapy for patients with cisplatin-eligible MIBC. Additionally, for cisplatin-ineligible MIBC patients, a trial could be designed with a combination of magrolimab and an immune checkpoint inhibitor. These potential treatment combinations warrant further exploration in order to advance the field of MIBC therapy.

## Figures and Tables

**Figure 1 jpm-13-00885-f001:**
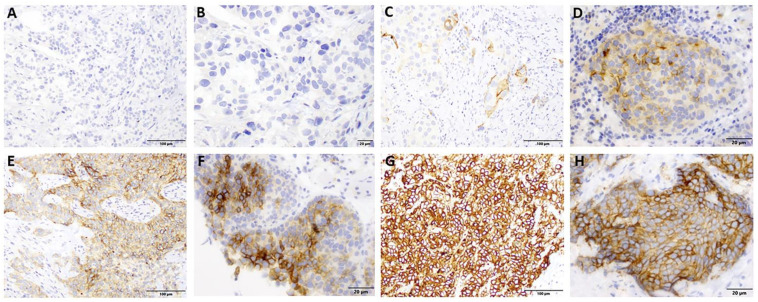
CD47 staining by IHC. (**A**,**B**) No staining at 20x and 40x magnification, respectively. (**C**,**D**) Weakly positive staining at 20x and 40x magnification, respectively. (**E**,**F**) Moderately positive staining at 20x and 40x magnification, respectively. (**G**,**H**) Strongly positive staining at 20 and 40x magnification, respectively.

**Figure 2 jpm-13-00885-f002:**
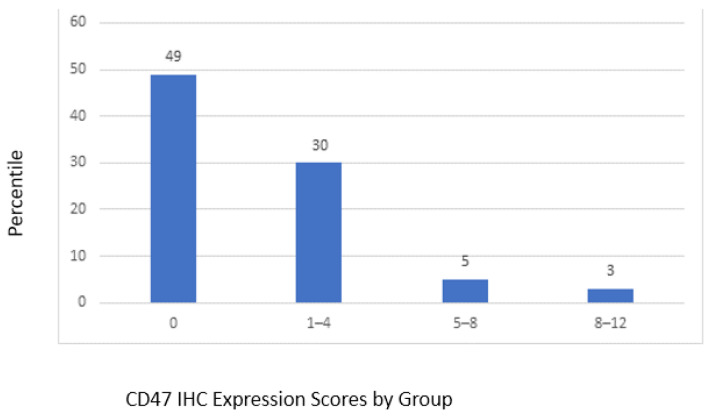
Distribution of CD47 IHC expression in TURBT specimens.

**Figure 3 jpm-13-00885-f003:**
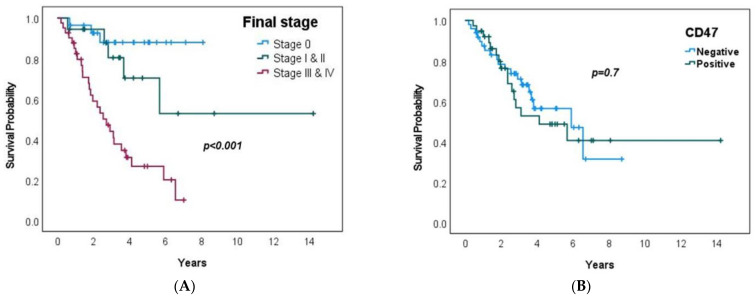
(**A**) (**left**) Kaplan–Meier curve of OS and CD47 negative or positive expression. (**B**) (**right**) Kaplan–Meier curve of OS and final staging.

**Figure 4 jpm-13-00885-f004:**
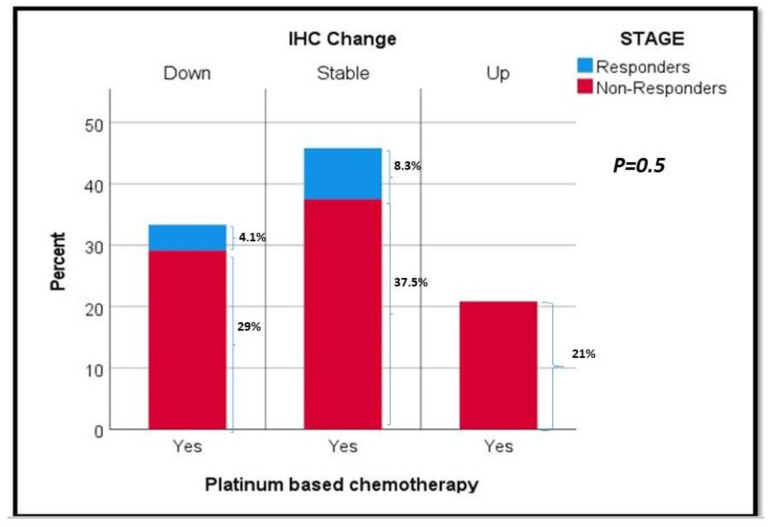
Distribution of CD47 IHC changes (TURBT-RC) and responders vs. non-responders with platinum-based chemotherapy.

**Table 1 jpm-13-00885-t001:** Association between CD47 IHC expression from TURBT specimens and clinicopathological characteristics.

Clinical Variables	All (n = 87)	CD47 IHC Negative Expression (n = 49)	CD47 IHC Positive Expression (n = 38)	*p*-Value
	% (n/N)	% (n/N)	% (n/N)	
Age (median range)	66 (39–84)	67 (43–84)	62 (39–84)	0.71
≤60	36.8 (32/87)	36.7 (18/49)	36.8 (14/38)	
≥60	63.2 (55/87)	63.3 (31/49)	63.2 (24/38)	
Sex				0.60
Female	20.7 (18/87)	18.4 (9/49)	23.7 (9.38)	
Male	79.3 (69/87)	81.6 (40/49)	76.3 (29/38)	
Race				0.72
Caucasian	95.4 (83/87)	96 (47/49)	95 (36/38)	
African American	2.3 (2/87)	2 (1/49)	2.5 (1/38)	
Asian	1.15 (1/87)	0 (0/49)	2.5 (1/38)	
Missing	1.15 (1/87)	2 (1/49)	0 (0/38)	
Neoadjuvant chemotherapy				0.32
Yes	75.9 (66/87)	71.4 (35/49)	18.4 (7/38)	
No	24.1 (21/87)	28.6 (14/49)	81.6 (7/38)	
Pathologic staging				0.83
Responders	35.6 (31/87)	36.7 (14/49)	34.2 (13/38)	
Non-responders	64.4 (56/87)	63.3 (35/49)	65.8 (13/38)	
Final pathologic stage				1.00
Stage 0 (ypT0N0 and ypTisN0)	32.2 (28/87)	32.7 (16/49)	31.6 (12/38)	
Stage 1 (ypT1N0)	1.2 (1/87)	2 (1/49)	0 (0/38)	
Stage 2 (ypT2N0)	19.5 (17/87)	18.4 (9/49)	21 (8/38)	
Stage 3 3a (≥ ypT3N0 or ypN1) 3b (any ypT yp ≥ N2)	42.5 (37/87)	42.9 (21/49)	42.1 (16/38)	
Stage 4a (ypT4b any ypN)	4.6 (4/87)	4 (2/49)	5.3 (2/38)	
Tumor recurrence (either local or distant recurrence)				1.00
Yes	30 (26/87)	32.7 (16/49)	29 (11/38)	
No	70 (61/87)	67.3 (33/49)	71 (27/38)	
Vital status				0.83
Alive	69 (60/87)	61.2 (30/49)	57.9 (22/38)	
Death	31 (27/87)	38.8 (19/49)	42.1 (16/38)	

**Table 2 jpm-13-00885-t002:** Univariate and multivariate analyses for PFS and OS.

Clinical Parameters		Progression Free Survival			Overall Survival		
Variable	Subset	HR (Hazard Ratio)	95% CI (Range)	*p*-Value	HR (Hazard Ratio)	95% CI (Range)	*p*-Value
UNIVARIATE ANALYSIS							
Gender	Male vs. Female	0.7	0.35–1.6	0.42	1.2	0.53–2.8	0.65
Age (year)	>60 vs. ≤60	2.9	1.3–6.3	0.009	3.2	1.4–7.3	0.006
CD47 IHC expression	Negative expression vs. Positive expression	1.0	0.5–2.0	0.94	1.1	0.6–2.2	0.7
Pathological stage	Responders vs. Non-responders	5.1	2.0–13.1	<0.001	5.1	1.8–14.5	0.002
Neoadjuvant chemotherapy	No vs. Yes	0.8	0.4–1.5	0.43	1.0	0.5–2.0	0.9
Final stage				<0.001			<0.001
	Stage 0	Ref.					
	Stage 1 & 2	2.6	0.7–9.4	0.13	2.5	0.6–10.4	0.22
	Stage 3 & 4a	8.1	2.8–23.3	<0.001	8.8	2.7–29	<0.001
MULTIVARIATE ANALYSIS							
Age (years)	>60 vs. ≤60	1.6	0.68–3.56	0.3	1.8	0.75–4.38	0.18
Pathological stage	Responders vs. non-responders	0.9	0.12–7.5	0.9	0.8	0.11–6.41	0.85
Final stage							<0.001
	Stage 0	Ref.					
	Stage 1 & 2	2.5	0.7–8.9	0.17	2.2	0.5–9.5	0.3
	Stage 3 & 4	10.2	3.54–30.2	<0.001	10.9	3.2–37	<0.001

**Table 3 jpm-13-00885-t003:** Association between CD47 IHC changes (TURBT-RC) and response to platinum-based chemotherapy.

Clinical Parameters		CD47 Immunohistochemistry Scoring (Total N = 40)				
Variable	Subset	Total (N)	Down (N)	Stable (N)	Up (N)	*p*-Value
Neoadjuvant chemotherapy						
No	Responders	1	0	1	0	0.8
	Non-Responders	15	4	10	1	
Yes	Responders	3	1	2	0	0.5
	Non-responders	21	7	9	5	

## Data Availability

Data will be available from the corresponding author on reasonable request.

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
