# Peer review of "Association of CD47 Expression with Clinicopathologic Characteristics and Survival Outcomes in Muscle Invasive Bladder Cancer"

_jpm, 2023, doi:10.3390/jpm13060885_

Round 1
Reviewer 1 Report
This is a good manuscript with good number of patients. I would like to continue and looking forward to seeing more patients, and hopefully, there will be some statistical significance.
I also would like to have some prospective trial design from this data if possible.
I'd like to add a short paragraph at end for conclusion/summary.
Author Response
Thank you for the comment. Authors added a conclusion and future direction for this as below.
onclusion:
CD47 expression is not a prognostic marker for MIBC patients. However, expression of CD47 was detected in nearly half of MIBCs, and future studies are needed to explore a potential role for anti-CD47 therapy in these patients. Furthermore, there was a slight positive trend between decreased CD47 levels (from TURBT to RC) in patients receiving NAC. More research is needed to understand how NAC may or may not modify immune surveillance mechanisms in MIBC.
Future Direction:
Kiss et al. studied the impact of macrophage-mediated phagocytosis and anti-tumor activity on bladder cancer cells (639V) and xenograft model 32. They investigated magrolimab (anti-CD47 therapy) as monotherapy and in combination with cytotoxic chemotherapy (gemcitabine-cisplatin) in both vivo and vitro 32. The study confirmed that combining magrolimab with cytotoxic chemotherapy resulted in synergistic effects, notably enhancing phagocytosis and reducing tumor growth compared to magrolimab monotherapy or chemotherapy alone 32. Currently, a phase I clinical trial (NCT05738161) is underway to explore the combination of magrolimab with cytotoxic chemotherapy in advanced urothelial carcinoma.
Lakhani et al. conducted a phase 1b study of combined magrolimab and avelumab (immune checkpoint inhibitor) in patients with immunotherapy naïve ovarian cancer who progressed within 6 months of prior platinum-based chemotherapy 33. The combination was well-tolerated with an acceptable safety profile and the recommended phase 2 dose for magrolimab was 45mg/kg once weekly 33. There was no reported dose-limiting toxicity. The most common treatment-related adverse events were headaches (62%), fatigue (47%), infusion-related reactions (44%), fever (38%), chills (35%), and nausea (35%). Among those 18 ovarian cancer pts included in the phase 1 study, 56% had stable disease, and 44% had disease progression 33.
Based on our own study and above findings, future clinical trials design could consider a potential neoadjuvant study that explores the combination of magrolimab and cytotoxic chemotherapy for patients with cisplatin-eligible MIBC. Additionally, for cisplatin-ineligible MIBC patients, designing with a combination of magrolimumab with immune checkpoint inhibitor. These potential treatment combinations warrant further exploration in order to advance the field of MIBC therapy.
Reviewer 2 Report
interesting article
Despite negative results and the authors decided to publish their work. Impressive and important decision.
The study is well performed and well presented.
minor revision:
IHC should be abbreviate in the simple summary section.
Author Response
Thank you for the comment. Authors abbreviated IHC in the abstract section.